# The Clinical Impact of Death Domain-Associated Protein and Holliday Junction Recognition Protein Expression in Cancer: Unmasking the Driving Forces of Neoplasia

**DOI:** 10.3390/cancers15215165

**Published:** 2023-10-26

**Authors:** Alexandros Pergaris, Ioannis Genaris, Ioanna E. Stergiou, Jerzy Klijanienko, Stavros P. Papadakos, Stamatios Theocharis

**Affiliations:** 1First Department of Pathology, Medical School, National and Kapodistrian University of Athens, 75 Mikras Asias Street, Bld 10, Goudi, 11527 Athens, Greece; alexperg@yahoo.com (A.P.); johngen30@gmail.com (I.G.); stavrospapadakos@gmail.com (S.P.P.); 2Department of Pathophysiology, Medical School, National and Kapodistrian University of Athens, 11527 Athens, Greece; stergiouioanna@hotmail.com; 3Department of Pathology, Curie Institute, 75005 Paris, France; jerzy.klijanienko@curie.fr

**Keywords:** DAXX, HJURP, biomarkers, cancer, prognosis

## Abstract

**Simple Summary:**

DAXX and HJURP are two proteins implicated in many physiologic processes and are considered important players contributing to the pathogenesis of a multitude of tumors. The aim of the present review of the literature is to retrieve and concisely present the data from studies conducted on human tissues that explored the expression of the aforementioned molecules in various tumor types. The researchers reported the tumor-promoting or tumor-suppressing properties of the two proteins, depending on the organ of origin, and correlated their expression with clinicopathological parameters. We report that enough data currently exist designating DAXX and HJURP as important factors in many tumors’ carcinogenesis and that both represent potential biomarkers for diagnosis, estimation of patients’ prognosis, therapy monitoring as well as targets for new therapeutic interventions.

**Abstract:**

Death domain-associated protein (DAXX) and Holliday junction recognition protein (HJURP) act as chaperones of H3 histone variants H3.3 and centromere protein A (CENPA), respectively, and are implicated in many physiological processes, including aging and epigenetic regulation, by controlling various genes’ transcription and subsequently protein expression. Research has highlighted both these biomolecules as participants in key procedures of tumorigenesis, including cell proliferation, chromosome instability, and oncogene expression. As cancer continues to exert a heavy impact on patients’ well-being and bears substantial socioeconomic ramifications, the discovery of novel biomarkers for timely disease detection, estimation of prognosis, and therapy monitoring remains of utmost importance. In the present review, we present data reported from studies investigating DAXX and HJURP expression, either on mRNA or protein level, in human tissue samples from various types of neoplasia. Of note, the expression of DAXX and HJURP has been associated with a multitude of clinicopathological parameters, including disease stage, tumor grade, patients’ overall and disease-free survival, as well as lymphovascular invasion. The data reveal the tumor-promoting properties of DAXX and HJURP in a number of organs as well as their potential use as diagnostic biomarkers and underline the important association between aberrations in their expression and patients’ prognosis, rendering them as possible targets of future, personalized and precise therapeutic interventions.

## 1. Introduction

With approximately 19 million new cases worldwide and an estimated 10 million cancer-related deaths in 2020, cancer represents one of the leading causes of mortality, heavily impacting patients’ quality of life and resulting in substantial socioeconomic ramifications [1]. Ongoing research continues to unveil the various molecular mechanisms behind oncogenesis. Complex pathways that encompass a multitude of molecules are implicated in a variety of processes that drive different types of tumors. Defining their role is considered of utmost importance, as it holds the key to the development of novel, personalized, and more effective therapeutic interventions. Epigenetic mechanisms, shaping the ever-changing landscape of a cell’s gene expression, are viewed as one of the main factors driving neoplasia. More specifically, the role of proteins implicated in DNA packaging and the proteins associated with them has become the research subject of a plethora of scientists who have attempted to shed light on their implications in gene expression and, ultimately, carcinogenesis. Among them, death domain-associated protein (DAXX) and Holliday junction recognition protein (HJURP) are two of the most extensively studied biomolecules due to their central contribution in a variety of epigenetic modifications implicated in neoplasia (Figure 1).

DAXX interacts with the H3.3/H4 dimer acting as a H3.3 histone chaperone [2,3]. Apart from its role as a H3.3 specific histone chaperone, it is also part of several molecular signaling pathways, including the FAS-DAXX-apoptosis signal-regulating kinase 1 (ASK1)-mitogen-activated protein 2 kinase (MAP2K) axis that induces apoptosis [4,5,6] and the DNA-damage-induced p53 activation [2]. HJURP acts as a centromere protein A (CENPA) histone chaperone, aiding its deposition during the late mitosis/early G1 phase of the cell cycle [7]. HJURP plays a vital role in safeguarding chromosome stability. On the other hand, its overexpression, induced by p53 mutations, is linked to ectopic deposition of CENP-A and tumor-promoting properties in various organs [7].

DAXX and HJURP are implicated in a multitude of cellular processes, including telomere lengthening, cell apoptosis, chromosome stability, and regulation of various oncogenes’ expression, all of which play key roles both in tumorigenesis and tumor suppression. Interestingly, DAXX and HJURP can either promote or hinder most of the aforementioned mechanisms, acting ultimately either as pro- or anti-tumorigenic factors, depending on the organ of origin or the specific type of tumor. Figure 2 is indicative of the diverse effects of DAXX and HJURP in tumorigenesis, as they can undertake a protective or aggravating role depending on the tissue of origin. DAXX acting either as a pro-apoptotic or an anti-apoptotic factor, contingent on the cell type and context, is a mere example of the complexity of these biomolecules’ role in cancer [8].

In the current review, we aim to document the studies investigating the association between DAXX and HJURP expression with human neoplasia of various tissues, as well as the possible correlations with clinicopathological parameters [9]. More specifically, we will try to decipher whether DAXX and HJURP are over- or under-expressed in solid tumors and hematologic neoplasms and whether the different expression patterns are linked to clinicopathological parameters, and we assess their utility as tools for diagnosis and prognostic biomarkers, as well as their possible application in therapy monitoring. We will also explore their role as potential targets for the development of novel, personalized therapeutic strategies. This review focuses on published research studying DAXX and HJURP expression either on mRNA or protein level in human tissue samples. Of note, we do not include studies conducted on non-human tissue specimens, xenograft models, and cell lines alone. The majority of the studies presented applied the methods of immunohistochemistry (IHC), IHC in tissue microarrays (TMA IHC), Western blot, reverse transcription polymerase chain reaction (RT-PCR), quantitative PCR (qPCR), next-generation sequencing (NGS), and whole-genome sequencing (WGS) to assess DAXX and HJURP protein or mRNA expression in malignant tissues and non-malignant control specimens. Researchers evaluated the presence or absence of expression, the expression patterns, the levels of expression and staining intensity along with the different expression patterns and protein localization (nuclear, cytoplasmic, or membranous). These results have been correlated with a number of clinicopathological parameters, including tumor histological grade, disease pathological stage, and patients’ disease-free survival (DFS) and overall survival (OS).

## 2. Head and Neck, Thymus, and Skin Tumors

### 2.1. Brain

Five studies in total investigated the impact of DAXX expression in brain tumors, often yielding contradictory results (Table 1). Cavalcante et al. reported the presence of significantly higher DAXX expression in grade 2 and grade 3 meningiomas compared to those classified as grade 1 [10]. Regarding glioblastoma multiforme (GBM) patients’ OS, alterations in DAXX genes associated with longer OS in the study conducted by Cantero et al. [11] and loss of DAXX expression has been linked to shorter OS by S. Ak Aksoy et al. [12]. On the other hand, DAXX mutated neuroblastoma (NBL) cases exhibited poorer outcome due to increased chemotherapy resistance [13]. Finally, alterations in DAXX expression often led to alternative lengthening of telomeres (ALT), a feature that in many cases bears important clinical impact. However, no significant differences in the clinicopathological characteristics of the group with abnormally shortened telomeres compared to the group with normal telomeres have been observed in pituitary adenoma cases [14].

HJURP seems to play an overall tumor-promoting role in brain carcinogenesis. In a large study that incorporated 267 malignant glioma specimens, HJURP overexpression correlated with poorer patients’ OS [15]. Similarly, two more studies reported the link between lower OS and HJURP upregulation in brain tumor cases [16,17].

In conclusion, DAXX exerts a tumorigenic role in brain carcinogenesis (Figure 2) and constitutes, furthermore, a potential diagnostic biomarker for tumor grade assessment, as well as an indicator of prognosis, specifically in the fields of patients’ OS and expected response to therapy (Figure 3). Similarly, HJURP exhibits tumor-promoting properties in brain tumorigenesis (Figure 2), with its expression linked to patients’ poorer OS (Figure 3).

### 2.2. Oral Cavity

Both DAXX and HJURP expression was reported upregulated in oral cancer tissue specimens compared to non-tumorous control samples [18,19], with HJURP overexpression exhibiting further association with patients’ age and shorter OS [19]. Therefore, DAXX and HJURP seem to promote carcinogenesis of the oral cavity (Figure 2), with HJURP level of expression exhibiting prognostic value as well, with its upregulation linked to grimmer outcomes (Figure 3).

### 2.3. Thymus

In thymic epithelial tumor (TET) patients, increased cytoplasmic HJURP expression associated with advanced Masaoka–Koga stage and its presence correlated with lymphocyte-poor TETs [20]. Further research is called for to investigate the potential tumor-promoting role of HJURP in TETs as well as its diagnostic value (Figure 3).

### 2.4. Skin

The only study exploring the role of HJURP in skin tumors concerned melanomas, included a large sample of 458 cases and underlined its tumorigenic properties, as not only was HJURP overexpressed in tumor tissues compared to normal ones, but it also revealed the link between its upregulation and shorter patients’ OS [21]. Therefore, HJURP appears to promote the development of skin melanocytic lesions (Figure 2) and to constitute a herald of poorer patients’ prognosis (Figure 3).

## 3. Lung, Breast, and Sarcomas

### 3.1. Lung

High DAXX expression has been associated with lower OS in lung adenocarcinoma (AC) patients, although not significantly (*p* = 0.216), while high concomitant DAXX and ATRX expression has been associated with better OS in these patients [22].

On the other hand, the role of HJURP in lung carcinogenesis has been more extensively investigated, with four studies reporting the association between its overexpression and lower patients’ OS [23,24,25,26], and two of them further correlating it with higher TNM disease stage [25,26].

It is, therefore, evident that more research is needed to shed light on the possible tumorigenic role of DAXX in lung AC (Figure 2 and Figure 3). Meanwhile, HJURP exhibits a clear negative effect in neoplastic disease of the lung, further serving as a potential biomarker for disease staging and patients’ prognosis (Figure 2 and Figure 3).

### 3.2. Breast

In a large study that included 220 breast cancer (BC) cases, DAXX expression correlated with tumor grade, necrosis, vascular invasion, and positive hormonal receptor status. Low levels of DAXX expression associated with lower DFS but not OS [27]. In solid papillary carcinoma (SPC) patients, loss of DAXX expression was significantly associated with lymphatic invasion [28].

Four studies in total underline the tumor-promoting role of HJURP in BC specimens. Bravaccini et al. reported a greater than sevenfold risk of relapse in patients highly expressing HJURP in the stroma of in situ breast carcinoma specimens [29]. Milioli et al. focused on the different patterns of HJURP expression in basal-like breast cancer (BLBC) cases, observing high levels of HJURP expression in basal II basal-like subtype (BL2) specimens [30,31]. This observation possibly has a significant impact on clinical decision making and planning of optimal therapeutic strategy, as BL1 and BL2 subtypes exhibit higher expression of cell cycle and DNA damage response genes and representative cell lines preferentially responded to cisplatin. Moreover, high HJURP mRNA levels were significantly associated with estrogen-receptor and progesterone-receptor negative status, advanced Scarff–Bloom–Richardson grade, younger age at diagnosis, Ki67 proliferation indices, and shorter DFS and OS [32]. Montes de Oca et al. also correlated high HJURP mRNA levels with shorter patients’ DFS along with lower distant metastasis-free survival, higher possibility of developing metastasis, and shorter OS in the luminal A subtype cases [33].

As a conclusion, DAXX appears to exert a protective role in BC (Figure 2), exhibiting value as a prognostic biomarker as well (Figure 3). On the other hand, HJURP negatively impacts neoplastic disease of this specific organ, potentially aiding in disease precise diagnosis and prognosis (Figure 2 and Figure 3).

### 3.3. Sarcomas

In a study that included 106 angiosarcoma specimens, all tissue samples retained DAXX immunoexpression, while no association between its levels and clinicopathological parameters was observed [34]. On the contrary, DAXX immunoexpression was observed as significantly higher in chondrosarcoma than in osteochondroma tissues, and positive DAXX expression positivity associated with shorter patients’ OS [35]. DAXX gene methylation has been detected in leiomyosarcoma (LMS) specimens [36] and, in a similar study, none of the 92 LMS cases lost DAXX immunoexpression [37]. In gastrointestinal stromal tumors (GISTs), only 3.3% of cases showed loss of DAXX immunoexpression and telomere dysregulation (defined as telomerase reverse transcriptase (TERT) promoter mutations or loss of either ATRX or DAXX expression) was frequently seen in GISTs of non-gastric origin but did not statistically correlate with any clinicopathological characteristics, DFS or OS [38].

Data on HJURP expression in human bone and soft tissue tumors remains limited, as only one study currently exists that investigated its levels in 58 synovial sarcoma (SS) cases. The scientists reported that HJURP gene is over-expressed in SS tissues, along with four other genes, namely non-SMC condensin I complex subunit G (NCAPG), Xenopus kinesin-like protein 2 (TPX2), CENPA, and NDC80 [39].

Therefore, DAXX role is designated as tumorigenic in a subset of sarcomas, namely chondrosarcomas, with a potential role as a biomarker of grim prognosis (Figure 2 and Figure 3). Similarly, HJURP appears to play a tumor-promoting role in SSs (Figure 2).

Table 2 summarizes the results of the studies carried out on DAXX and HJURP expression in lung and breast tumors as well as sarcomas.

## 4. Gastrointestinal Tract, Liver, Biliary Tract, and Pancreas

### 4.1. Esophagus

Ko et al. examined DAXX immune expression in 60 esophageal squamous cell carcinoma (ESCC) specimens using adjacent non–tumorous tissue samples as controls and reported that while DAXX immunostaining remained negative or weak in normal tissues, in cancerous samples it associated with lymph node metastasis, American Joint Committee on Cancer (AJCC) stage, distant metastasis and OS in patients with stage IIB and stage IV disease [40]. In accordance with these results are the ones observed by Yuen et al., who correlated DAXX immunoexpression with nuclear DJ-q expression. The latter was significantly higher in ESCC and lymph node metastases tissues and associated with shorter OS and higher tumors’ metastatic potential [41].

Overall, DAXX upregulation is designated as a tumor-promoting factor in ESCCs (Figure 2) and could be utilized in the fields of disease staging, tumor grading, and assessment of OS (Figure 3).

### 4.2. Stomach

Chen et al. stressed the importance of DAXX protein subcellular localization in neoplastic tissues. Interestingly, in their study that incorporated 323 gastric cancer (GC) cases, cytoplasmic DAXX expression was associated with a longer OS and DFS, while high nuclear DAXX expression suggested a poorer OS [42]. Similarly, one more study highlighted that it was the DAXX Nuclear/Cytoplasmic ratio (NCR) that was associated with cancerous tissues. More specifically, in normal gastric mucosal cells, DAXX was expressed in the cytoplasm but was absent in the nucleus. Cells with intestinal metaplasia appeared to express DAXX in both the nucleus and the cytoplasm, and in tumor cells, DAXX exhibited only nuclear expression [43]. Moreover, a high DAXX NCR ratio was associated with shorter DFS and OS.

In conclusion, nuclear DAXX overexpression exerts a tumorigenic role in GC (Figure 2), while it also associates with poorer overall prognosis (Figure 3).

### 4.3. Colon

The role of DAXX in the field of colon carcinogenesis appears to be controversial, as 3 studies attribute tumor-suppressive properties to DAXX [44,45,46] while one group of researchers report it as a tumorigenic factor [47]. DAXX expression was found significantly lower in colorectal carcinoma (CRC) cases with positive serum carcinoembryonic antigen (CEA) screening results (defined as serum CEA levels of >5 ng/mL) compared to patients with negative CEA screening levels. Moreover, DAXX expression was significantly correlated with CD24 expression. Specifically, reduced DAXX expression was associated with reduced CD24 expression in CRC samples [44]. DAXX might be considered a potential regulator of CD24 or β-catenin expression, which may be correlated with the proliferative and metastatic potential of CRC. A significant increase in proliferation was observed in DAXX-knockdown CRC cells compared with the negative controls. DAXX knockdown significantly increased 293 T and Hct116 cell migration, suggesting that, indeed, DAXX may play a protective role in CRC [44]. Similarly, Liu et al. explored the expression of DAXX in 8 pairs of matched primary CRC tissue and liver metastatic CRC tissue samples, reporting it lower in liver metastases than in primary CRC tissues [45]. These results suggest a possible association between DAXX downregulation and CRC metastasis. Additionally, Tzeng et al. observed reduced DAXX expression in CRC tissues compared to normal colon tissues [46]. On the other hand, Huang et al., utilizing both IHC and qRT-PCR, reported higher DAXX expression in CRC specimens compared to non–neoplastic ones [47]. It is therefore clear that further studies are required to elucidate the complex effects of DAXX in colon carcinogenesis.

Interestingly, HJURP also seems to act as a tumor suppressor in this specific organ. Kang et al. conducted immunohistochemical studies in 162 CRC cases and observed a beneficial role of high HJURP expression in tissues to CRC patients’ OS [48].

To conclude, DAXX has been reported as exerting both tumor-promoting and tumor-suppressing properties in CRC (Figure 2), while HJURP appears to play a protective role and was correlated with longer OS (Figure 2 and Figure 3).

### 4.4. Liver

Retrieving data from the cancer genome atlas (TCGA) and the Gene Expression Omnibus (GEO) datasets, Ding et al. reported that the mRNA expression of 9 genes in total, namely enhancer of zeste homolog 2 (*EZH2*), cyclin-dependent kinase 1 (*CDK1*), *CENPA*, RD54-like (*RAD54L*), PDZ binding kinase (*PBK*), helicase, lymphoid specific (*HELLS*), *HJURP*, aurora kinase A (*AURKA*), and aurora kinase B (*AURKB*), was associated with hepatocellular carcinoma (HCC) patients’ OS as well as invasion depth of tumor and distant metastasis [49]. In accordance with these findings, another group of researchers observed higher HJURP expression in HCC tissues compared to adjacent normal ones as well as HJURP upregulation in tissues of patients with advanced HCCs compared to those of patients with early-stage disease. Moreover, HJURP overexpression was associated with poor patients’ DFS and OS [50]. Similarly, Hu et al. investigated HJURP mRNA and protein expression through the methods of real-time PCR and IHC, reporting that high HJURP expression was significantly associated with tumor size (tumors >5 cm), Barcelona clinic liver cancer stage, tumor number, tumor differentiation, TNM stage and poorer patients’ OS [51].

It is evident that HJURP enhances liver neoplastic process (Figure 2) and can potentially aid in disease TNM staging, tumor grading as well as serve as a prognostic marker for patients’ DFS and OS (Figure 3).

### 4.5. Biliary Tract

The role of HJURP in biliary tract neoplasia was investigated by Yang et al. who reported increased HJURP expression in cholangiocarcinoma (CCA) tissue samples compared to non-tumorous ones. Additionally, HJURP overexpression was associated with lower OS in cases of intrahepatic and perihilar CCA but not in patients with distal CCA [52]. Thus, the study highlights the potential role of HJURP as a tumor-promoting factor (Figure 2) and a marker of worse prognosis (Figure 3) in CCAs.

### 4.6. Pancreas

The contribution of DAXX in pancreatic neuroendocrine tumors (PanNETs) is undoubtedly among the most extensively studied. While it has been the subject of a considerable number of studies, no contradictory results have been reported, with researchers agreeing on the overall tumor-promoting role of DAXX loss in pancreas tumorigenesis [53,54,55,56,57,58,59,60,61,62,63,64,65,66,67,68,69].

Jiao et al., based on a sample comprising 68 sporadic PanNET specimens, detected mutations of *DAXX* genes in 12 out of 68 cases, which were associated with better patients’ OS [53]. One possible explanation for the difference in survival is that mutations in multiple endocrine neoplasia type 1 (*MEN1*) and *DAXX*/*ATRX* identify a biologically specific subgroup of PanNETs. Another group of researchers conducted IHC studies in large sample comprising 1322 neuroendocrine tumors (NETs) in total that included 561 primary nonfunctional PanNETs (NF-PanNETs), 107 NF-PanNET metastases and 654 primary, non-pancreatic non-functional NETs and NET metastases. Loss of DAXX was correlated with the presence of distant metastases/recurrence and DFS rates were shorter in patients with small NF-PanNETs (≤2.0 cm) with DAXX loss. Moreover, only a small subset of pulmonary carcinoids exhibited DAXX loss. Therefore (loss of) DAXX emerges a potential highly specific diagnostic biomarker to indicate the possibility of a pancreatic primary tumor [54]. Ueda et al. carried out IHC studies in 44 PanNET specimens and observed that low DAXX expression correlated with nonfunctional tumors, higher Ki-67 index, World Health Organization (WHO) grade G2 tumors, higher recurrence rate, worse patients’ DFS and presence of synchronous liver metastasis, although insignificantly (*p* = 0.081) [55]. Interestingly, in a study that incorporated 13 pancreatic neuroendocrine neoplasm (NEN) specimens (4 G1, 5 G2, and 4 G3 tumor samples), DAXX exhibited lower mRNA expression in the tumor relative to that in the non-tumor tissue in G1 and G2 specimens. In contrast, G3 samples exhibited higher DAXX mRNA expression in tumor tissues relative to non-tumorous ones. Additionally, DAXX IHC staining was absent in non-tumor, G1 and G2 tumor tissues, while strong nuclear staining was observed in less than 25% of cells in G3 tumor tissues [56]. DAXX mutation was identified in 13 out of 51 cases (25%) of 51 neuroendocrine liver metastases specimens, and the presence of DAXX mutation was associated with shorter hepatic progression-free survival (PFS) [57]. Park et al. reported that loss of DAXX protein expression was associated with poor patients’ OS. However, in the subgroup of cases with metastatic PanNETs, it was correlated with longer OS rates [58]. This observation is in accordance with the one made by Kim et al., who also reported that although loss of ATRX/DAXX expression was associated with reduced DFS, in patients with distant metastases, the loss of ATRX/DAXX expression was associated with better OS [65]. The link between loss of DAXX expression and DAXX mutations with lower patients’ DFS or OS was highlighted by three more studies [59,60,61]. On the other hand, in PanNET specimens from patients with Von Hippel-Lindau (VHL) syndrome, no statistically significant difference in tumor DAXX protein expression by pathologic diagnosis, clinical data, VHL exon mutation, and functional imaging results was identified [62]. Additionally, the loss of DAXX expression was correlated not only with shorter DFS rates [63] but also with postoperative hepatic recurrence [63] and an increased risk of metastasis [64]. Likewise, the loss of DAXX IHC expression was observed in 64 out of 100 cases of grade 1 and 2 primary PanNET specimens [66] and 16/105 PanNET specimens in a similar study [67] which reported that while the loss of DAXX expression alone did not significantly correlate with poor patients’ OS, the loss of either DAXX or ATRX was associated with higher Ki67 proliferative index, high tumor grade, nodal involvement, higher disease stage, infiltrative borders, and lymphovascular invasion. Similarly, the loss of ATRX/DAXX immunohistochemical expression was identified in 18/46 cases of PanNET tumors and was correlated with tumor size larger than 5 cm [68]. Lastly, Chen et al. examined not only NET specimens but also pancreatic adenocarcinoma (PDAC) samples in their study. The loss of DAXX expression was detected in 29.9% of all NET cases but not in any of the non-neoplastic pancreatic tissues or PDACs and was overall associated with Ki-67 index and higher histologic tumor grades [69].

In pancreatic tumors, HJURP expression has only been investigated in PDAC specimens and was observed significantly higher in cancerous tissues than in adjacent normal ones. Furthermore, it was linked to poor patients’ OS [70].

In conclusion, DAXX exerts an undisputed protective role in pancreatic NET tumorigenesis, with DAXX loss and ALT comprising the driving force behind this type of neoplastic process (Figure 2). Moreover, researchers have underlined the utility of DAXX as a diagnostic biomarker for assessing disease TNM stage and tumor grade as well as predicting patients’ DFS and OS (Figure 3). On the contrary, HJURP exhibits a tumorigenic role in PDAC (Figure 2) and has been linked to worse disease outcome (Figure 3).

The data from studies conducted investigating the role of DAXX and HJURP in the gastrointestinal tract, liver, biliary tract, and pancreas carcinogenesis are presented in Table 3.

## 5. Urinary Tract, Prostate, Testis, and Adrenal Glands

### 5.1. Kidney

DAXX loss of expression, although almost always observed in studies investigating PanNET specimens, was absent in primary renal NET cases [71].

Researchers who conducted studies on HJURP expression in clear cell renal cell carcinoma (ccRCC) specimens reported contradictory results regarding its role in tumorigenesis. More specifically, Wei et al. associated its overexpression with reduced patients’ DFS and OS [72]. On the other hand, one study observed no association between HJURP expression and clinicopathological parameters [73] and another detected lower HJURP expression in RCC tissues compared with that in the adjacent paracancerous ones [74]. In contrast, Zhang et al. reported it as upregulated in ccRCC tissues compared to normal ones and further associated HJURP expression with TNM stage, histopathological grade, and poorer patients’ OS [75]. Moreover, HJURP levels of expression were correlated with infiltration of various immune cells and expression of a wide range of immunocyte gene markers, showing that HJURP plays an essential role in the tumor microenvironment by regulating immunocyte infiltration [75].

Overall, one study investigated through qRT-PCR and Western blot the expression of HJURP and reported it was lower in ccRCC tissues compared with that in the adjacent paracancerous ones [74]. Another study, based on the data collected from TCGA database from 52 ccRCC samples, observed no association between HJURP expression and clinicopathological parameters [73]. Finally, two studies that also retrieved data from TCGA database and incorporated 955 ccRCC tumor specimens in total attributed tumorigenic properties to HJURP overexpression [72,75]. Therefore, it is possible that aberrations in HJURP expression are implicated in kidney carcinogenesis, which include both down- and upregulated expression in tumor tissues.

Therefore, DAXX does not seem to participate in renal carcinogenesis, with the role of HJURP still remaining unclear as far as neoplastic processes of the kidney are concerned (Figure 2).

### 5.2. Bladder

DAXX IHC expression was observed in 90% of urothelial carcinoma (UC) nuclei compared to 70% of normal urothelium nuclei. Moreover, in UC tissues, nuclei adjacent to the stroma exhibited slightly lower expression than in the intermediate cell layers, whereas in normal urothelium tissues, expression was slightly greater compared to the intermediate and superficial cell layers [76]. Based on their observations, the authors suggest the presence of altered protein expression of DAXX in UC and in its preinvasive phases. In small cell bladder carcinoma specimens, the loss of DAXX expression was detected in five (16.1%) cases, including four patients who presented with limited stage disease. All four patients who presented with limited-stage disease underwent chemotherapy and remained alive during follow-up, implying that loss of DAXX may sensitize tumor cells to chemotherapy [77].

Therefore, it is evident that DAXX exerts an overall tumorigenic role in bladder cancer (Figure 2).

### 5.3. Prostate

Kwan et al. reported higher DAXX expression in prostate cancer tissues and DAXX overexpression was associated with tumor Gleason score and the presence of metastasis [78]. Likewise, DAXX expression was observed as significantly increased compared to both normal adjacent tissues and benign prostate hyperplasia samples by Jamali et al. [79]. Another study found that high DAXX expression correlated with transmembrane protease, serine 2 (TMPRSS2)/ERG rearrangement, ERG expression, high tumor Gleason grade, advanced pT stage, high Ki-67 labeling index, and early prostate-specific antigen (PSA) recurrence [80]. Lastly, Lorena et al. reported DAXX mRNA levels as increased in metastatic prostate cancer tissues but low in normal prostate glands, and increased DAXX expression was associated with higher tumor Gleason scores and lower patients’ OS [81].

HJURP was also reported as overexpressed in prostate cancer tissues compared with benign prostate tissues, and high HJURP expression was associated with positive PSA levels, high Gleason score, advanced pathological stage, presence of metastasis, PSA failure, and shorter biochemical recurrence-free survival [82]. Luo et al. also observed a link between the lower biochemical recurrence-free survival and high HJURP expression [83].

In conclusion, both DAXX and HJURP are implicated in prostate carcinogenesis (Figure 2), with their overexpression appearing to hold value for disease staging and tumor grading as well as patients’ prognosis, in the case of DAXX (Figure 3).

### 5.4. Adrenal Glands

Mete et al. reported that the global loss of DAXX expression was associated with longer patients’ DFS in adrenal cortical carcinoma cases, suggesting a possible tumorigenic role of the aforementioned molecule in adrenal gland carcinogenesis [84] (Figure 2).

Table 4 summarizes the results from studies investigating DAXX and HJURP expression in tumors of the urinary tract, prostate, testis, and adrenal glands.

## 6. Gynecologic Tumors

Table 5 summarizes the data reported in the following sections.

### 6.1. Ovary

Pan et al. reported increased DAXX expression across a wide spectrum of ovarian tumors, including cystadenomas, endometrioid carcinomas, clear cell carcinomas, and mucinous cystadenomas. On the contrary, granular cell tumor samples as well as normal ovary tissue specimens exhibited weak DAXX immunostaining [8]. In serous effusion specimens, DAXX expression in high-grade serous carcinoma cells was associated with poor patients’ OS [85]. On the other hand, in a study that included 187 epithelial ovarian cancer specimens, no statistically significant association between DAXX mRNA expression and clinicopathological parameters was observed [86].

HJURP immunoexpression in ovarian cancer cases has been investigated in two studies, both of which underlined its tumor-promoting properties. The first study reported an association between increased HJURP expression and high (≥200 U/mL) serum carbohydrate antigen 125 (CA125) levels, large (≥3000 mL) ascites volume, the presence of metastasis in the omentum/peritoneum, as well as shorter patients’ DFS and OS [87]. Li et al. also observed that high HJURP expression levels were linked to lymph node metastases and lower OS [88].

Therefore, both DAXX and HJURP are shown to participate in ovarian neoplastic processes (Figure 2), with further implications for patients’ OS, as far as DAXX is concerned, and disease staging as well as for patients’ DFS and OS in the case of HJURP (Figure 3).

### 6.2. Uterus

DAXX homozygous deletions were detected in 11% of endometrial clear cell carcinoma cases [89]. In leiomyosarcoma patients, loss of ATRX or DAXX expression was associated with lower DFS and lower OS [90], while only one case of cellular uterine leiomyoma (UL) exhibited the loss of DAXX expression [91].

Regarding HJURP expression, increased HJURP mRNA expression in endometrial carcinoma specimens was linked to high tumor grade, disease recurrence, and worse patients’ OS [92].

DAXX appears to play a protective role in uterine carcinogenesis (Figure 2) with potential utility as a prognostic biomarker (Figure 3). On the contrary, HJURP acts as a tumor-promoter in this specific organ (Figure 2), with possible implications in disease diagnosis and prognosis (Figure 3).

### 6.3. Cervix

In cervical neoplastic tissues, DAXX intracellular localization seems to be implicated in different stages of disease progression. DAXX IHC expression was located in the nucleus of normal cervical epithelial cells, in the nuclear membrane and, to a lesser extent, in the nucleus of cervical intraepithelial neoplasia grade 1 (CIN1) cells and in the cytoplasm and cell membrane in cervical intraepithelial neoplasia grade 2 (CIN2), CIN3, and cervical cancer cells [93]. It is possible that during the progress of cervical cancer, DAXX gradually translocates from the nucleus into the nuclear membrane, cytoplasm, and cell membrane. DAXX is eventually located in the cytoplasm and cell membrane in CIN2, CIN3, and cervical cancer cells.

As is the case for gastric cancer, DAXX intracellular location seems to constitute a key factor regarding its tumorigenic role in cervical cancer (Figure 2).

## 7. Hematological Malignancies

Despite the quite extensive research conducted in the field of solid tumors, few studies investigate DAXX and HJURP expression and its clinical correlations in the setting of hematological malignancies. In this section, given the complexity of the classification of hematological neoplasms, we present them subdivided into main disease categories (Table 6).

### 7.1. Acute Leukemias

For acute leukemias (ALs), most of the published research focuses on the pediatric population. Three Chinese studies investigated DAXX expression via IHC assays in pediatric ALs. Jingqiao et al. reported a statistically significant increased rate of DAXX protein expression in all leukemia subgroups studied, namely 30.38% for acute lymphoblastic leukemia (ALL), 66.67% for acute non-lymphocytic leukemia (ANLL), and 46.67% for acute myeloid leukemia (AML) compared to 6.67% for healthy controls [94]. In accordance with this, Liu et al. documented a significantly higher DAXX protein expression rate of 38% overall for pediatric ALs compared to a 5% rate for the control group, also highlighting that children diagnosed with ANLL had significantly higher DAXX expression levels (62.5%) than those diagnosed with ALL (26.5%). Analyzing the data based on the leukemia subtype, increased DAXX expression was statistically significant only for the ANLL group and not for the ALL compared to controls. Given this finding, the ALL group was further divided in two subgroups, high-risk (HR) ALL and standard-risk (SR) ALL. Interestingly, DAXX protein was expressed in 55.6% of the HR ALL subgroup, but it was not expressed in the SR ALL subgroup [95]. Taking a step further, Han-hua et al. investigated the possible correlation between DAXX and nuclear factor kappa B (NF-κB) expression in pediatric acute leukemias. For ALL, DAXX expression was positively correlated with NF-κB expression. HR ALL was characterized by higher expression rates of both DAXX and NF-κB (55.6% and 77.8%, respectively) compared to SR ALL (0.0% and 16.7%, respectively) [96].

Regarding adult acute leukemia, only one study by Ding et al., applying deep sequencing to detect somatic mutations, reported DAXX to be among novel, recurrently mutated genes in relapsed AML [97].

We can therefore conclude that DAXX exerts a possible tumor-promoting role of DAXX in the setting of pediatric AL, specifically ANLL and HR ALL (Figure 2), and could serve as a prognostic marker for relapse in adult AML (Figure 3).

### 7.2. Myelodysplastic Neoplasms

Attieh et al. evaluated bone marrow (BM) aspiration samples from 77 patients with myelodysplastic neoplasms (MDS) and identified that DAXX mRNA expression in MDS CD34+ cells, assessed by real-time PCR, was significantly upregulated compared to healthy controls. Nevertheless, upregulated DAXX mRNA expression levels did not significantly correlate either with WHO or cytogenetic subset or with patient outcome [98].

Hence, even though further research is undoubtedly required, based on this single published study regarding MDS, a possible protumorigenic role of DAXX could be proposed (Figure 2).

### 7.3. Chronic Lymphocytic Leukemia

The only published report for DAXX protein expression in chronic lymphocytic leukemia (CLL) can be found in the study of Bojarska-Junak et al., where the authors documented a positive correlation between the expression of prostate apoptosis response-4 (Par-4) protein and DAXX protein, assessed via flow cytometry in peripheral blood and BM samples from patients with untreated CLL [99].

Given the correlation of Par-4 expression with adverse prognostic markers for CLL [99], we could assume an indirect association between DAXX expression and dismal CLL prognosis (Figure 3), a remark definitely requiring validation.

### 7.4. Lymphomas

Hornviller et al. assessed via Western blot DAXX protein expression in purified B cells derived from cell suspensions of lymph nodes from patients with diffuse large B cell lymphoma (DLBCL), identifying increased expression in the DLBCL cases compared to B cells derived from tonsils. The researchers also applied tissue microarray analysis to assess DAXX expression in 196 DLBCL tissue samples so that they could make comparisons with other types of lymphoproliferative diseases and correlations with disease outcomes, given the large scale of the sample size. DAXX protein expression was significantly increased in DLBCL compared to follicular lymphoma (FL) and CLL. A Kaplan–Meier analysis showed that high DAXX expression correlated with poor OS for all subtypes of DLBCL [100].

For Natural Killer/T Cell Lymphoma (NKTCL), IHC staining demonstrated that the defined by transcriptomic analysis HEA subtype (based on HDAC9, EP300, and ARID1A mutations) was characterized by DAXX protein overexpression, in IHC analysis, proposing that DAXX could serve as a hallmark for this molecular subtype. Of note, DAXX was significantly correlated with NF-κB and T cell receptor (TCR) pathways [101].

Considering the multiple lymphoma subtypes, the reports for the two aforementioned studies are less than adequate to allow conclusions. We can though underline a possible tumor-promoting contribution of DAXX in DLBCL (Figure 2), in the setting of which DAXX expression could be proposed as a new prognostic marker for the identification of patients with worse outcomes (Figure 3). For the less frequent cases of NKTCL DAXX could serve as a marker for molecular subtype discrimination.

### 7.5. Multiple Myeloma

Jia et al. performed in silico analyses on publicly available Gene Expression Omnibus datasets (GSE2113 and GSE6477), containing samples from patients with monoclonal gammopathy of undetermined significance (MGUS), multiple myeloma (MM), and plasma cell leukemia (PCL). A gradual increase in HJURP expression was observed along with disease progression from MGUS to PCL. Of note, relapsed MM cases demonstrated higher *HJURP* expression levels compared to non-relapsed. In the same study, a Kaplan–Meier analysis of several myeloma patient datasets (CoMMpass, HOVON, and UAMS) showed high HJURP value was associated with worse OS and progression-free survival (PFS) [102].

Thus, HJURP expression promotes disease progression in the setting plasma cell dyscrasias (Figure 2), while it can also be used as a prognostic marker of patients’ outcome (Figure 3).

## 8. Conclusions

Epigenetic modifications have been in the spotlight of cancer research in recent years, as a deeper understanding of their implication in tumorigenesis holds the key to early detection and precise diagnosis of tumors, the estimation of patients’ prognosis, and the development of novel, personalized, and ever more effective therapeutic interventions. DNA and RNA methylation mechanisms, histone modifications, and expression of non-coding RNAs constitute fundamental processes behind carcinogenesis and tumor-suppression [103,104,105,106,107,108,109]. Extensive research has emphasized the role of DAXX and HJURP, among other biomolecules implicated in epigenetic mechanisms, in the pathogenesis of a great variety of neoplasms. Their significance in crucial steps of carcinogenesis renders them potential biomarkers for timely disease diagnosis, assessment of patients’ prognosis, therapy monitoring as well as targets for the development of new, personalized treatment options. In PanNETs, DAXX is already routinely used as a trustworthy biomarker for the diagnosis of a well-differentiated neuroendocrine tumor as well as a predictor of a poor outcome. Researchers have revealed that ALT, driven by DAXX mutations, is the driving force behind PanNET neoplastic process, a mechanism separate from the molecular background of pancreatic neuroendocrine carcinoma tumorigenesis, which includes aberrations in the expression of p53 and retinoblastoma protein [110]. Apart from this established role in PanNet diagnosis and prognosis, a substantial amount of accumulating data highlight DAXX and HJURP as having either reliable tumor-promoting or tumor-suppressing properties and reveal their possible role as indicators of favorable or grim disease prognosis in various neoplastic diseases. More specifically, as presented in Table 1, Table 2, Table 3, Table 4, Table 5 and Table 6, researchers have linked DAXX and HJURP with a multitude of clinicopathological parameters, depending on the tumor tissue of origin (Figure 3).

Overall, an extensive review of the existing literature reveals an overwhelmingly adverse effect of DAXX and HJURP in carcinogenesis, with numerous harmful processes instigated by their overexpression in neoplastic cells. Ample data regarding the implication of the two biomolecules in tumors originating from the brain, lungs, esophagus, liver, prostate, and ovaries currently exist, with unequivocally damning evidence of their tumorigenic role. Therefore, it is becoming increasingly clear that DAXX and HJURP represent promising molecular targets of future anti-cancer medications, especially in the aforementioned organs of origin. For other types of neoplasms, such as those of the hematopoietic system, a limited number of studies support the protumorigenic role of DAXX and HJURP, rendering further investigations necessary to establish their function and possible application in the clinical setting. Nevertheless, aberrations in DAXX and HJURP expression are consistently linked to patients’ DFS and OS for cancers originating from the brain, lungs, breast, esophagus, liver, prostate, ovaries, the uterus. More studies are needed to help establish a stronger link between DAXX and HJURP expression and patients’ prognosis, in order for them to emerge as solid biomarkers of disease outcome. Lastly, as far as diagnosis is concerned, DAXX is already used for the identification of PanNETs. Few data exist designating them as useful for disease diagnosis, with further research needed to elucidate their potential role as trustworthy markers in that field.

The role of the tumor microenvironment in oncogenesis is increasingly in the epicenter of cancer research, as interplay between tumor cells and stromal elements is considered a defining factor for disease progression. Bravaccini et al. underlined that in in situ breast carcinoma cases, a greater than sevenfold risk of relapse was observed in patients highly expressing HJURP in stroma [29]. Moreover, immune reaction is emerging as a crucial element in carcinogenesis and especially patients’ prognosis. HJURP was highlighted as a regulator in this procedure in ccRCC specimens, with increased HJURP expression levels being associated with infiltration of different types of immune cells [75].

However, in many tumors, contradictory data have been reported regarding the properties of DAXX and HJURP in neoplasia. In CRC cases, different groups of researchers have attributed both tumor-suppressive [44,45,46] and tumor-promoting [47] properties to DAXX. Similarly, in ccRCC specimens, HJURP has been reported as having a negative [72,75], neutral [73], and protective [74] role in tumorigenesis. This can be attributed to the aberrant amplification of different molecular pathways leading to carcinogenesis. Each tumor carries a unique molecular signature, and a deeper understanding of it is a key step for the development of personalized therapeutic interventions. Nevertheless, such discrepancies render evident the need for further research to elucidate the complex role of DAXX and HJURP in carcinogenesis. Especially for tumors of the same histological type, where some research teams detected tumor-promoting and others tumor-suppressing properties of the aforementioned molecules, the investigation of their expression and its impact in large series of tumor samples is called for.

To summarize, an increasing amount of research in numerous types of neoplasia has underlined the implications of DAXX and HJURP in disease pathogenesis and has highlighted clinicopathological and prognostic correlations. Undoubtedly, additional studies are indispensable for the validation of these findings and for the establishment of these biomolecules as diagnostic and prognostic markers in cancer. Deciphering their intricate role in carcinogenesis will offer new therapeutic targets for personalized treatment approaches.

## Figures and Tables

**Figure 1 cancers-15-05165-f001:**
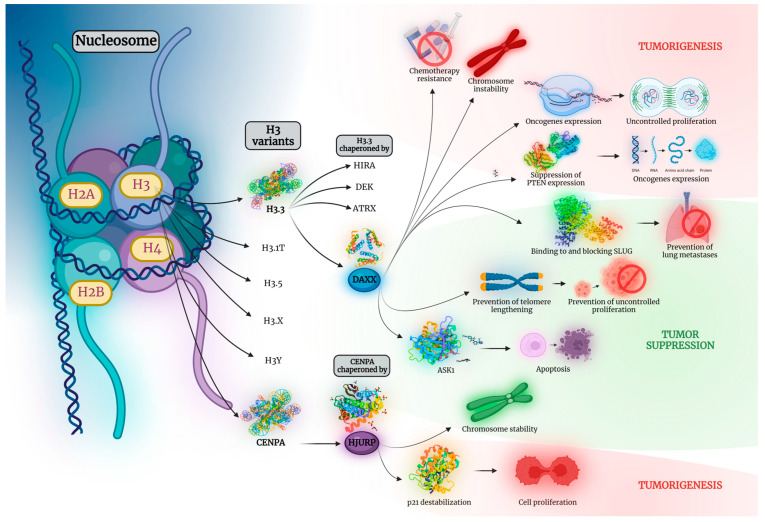
A nucleosome comprises approximately 146 base pairs of DNA wrapped around an octamer of Histones. Two each of the histones H2A, H2B, H3, and H4 constitute a histone octamer. H3 variants include H3.3, H3.1T, H3.5, H3.X, H3Y, and CENPA. H3.3 is chaperoned by 4 molecules, namely HIRA, DEK, ATRX, and DAXX. CENPA is in turn chaperoned by HJURP. Both DAXX and HJURP are implicated in a plethora of processes in tumorigenesis. To name but a few, DAXX overexpression is associated with resistance to chemotherapeutic regimens, chromosome instability, and upregulation of a multitude of oncogenes, either directly or by downregulating the tumor-suppressor gene PTEN, contributing to uncontrolled cell proliferation. Likewise, HJURP upregulation is proven to promote p21 destabilization, a protein implicated in cell cycle arrest, leading, thus, to uncontrolled cell proliferation as well. On the other hand, both DAXX and HJURP are shown to also hold tumor-suppressive properties. DAXX binds and blocks SLUG, an action linked to prevention of lung metastasis. Moreover, DAXX upregulation is associated with prevention of telomere lengthening and, therefore, restriction of cell proliferation. Moreover, it has been linked to ASK1 upregulation, a promoter of cell apoptosis. Similarly, HJURP is proven to promote chromosome stability. ASK1, apoptosis signal-regulating kinase 1; ATRX, alpha-thalassemia/mental retardation, X-linked; CENPA, centromere protein A; DAXX, death domain-associated protein; HIRA, histone regulator A; HJURP, Holliday junction recognition protein; PTEN, phosphatase and tensin homolog. Created with BioRender.com.

**Figure 2 cancers-15-05165-f002:**
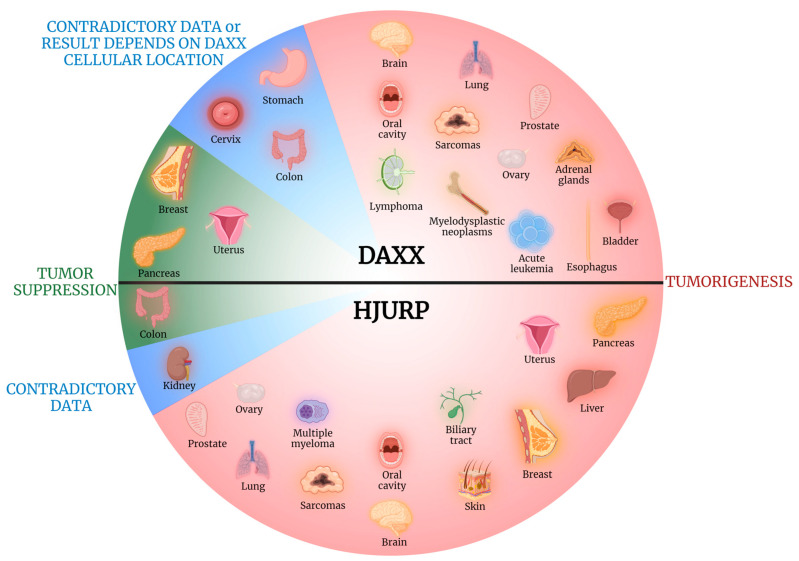
DAXX and HJURP have been reported as having both tumor-suppressive and tumor-promoting properties, depending on the organ studied. DAXX/HJURP exert tumor-suppressing roles on carcinogenesis of the organs depicted on the green part of the diagram and tumorigenic properties on the organs on the red part. Blue parts depict the organs where either contradicting data have been reported, concerning the tumorigenic or tumor-suppressive result of upregulation of the 2 biomolecules, or the type of action depends on their subcellular location. Created with BioRender.com.

**Figure 3 cancers-15-05165-f003:**
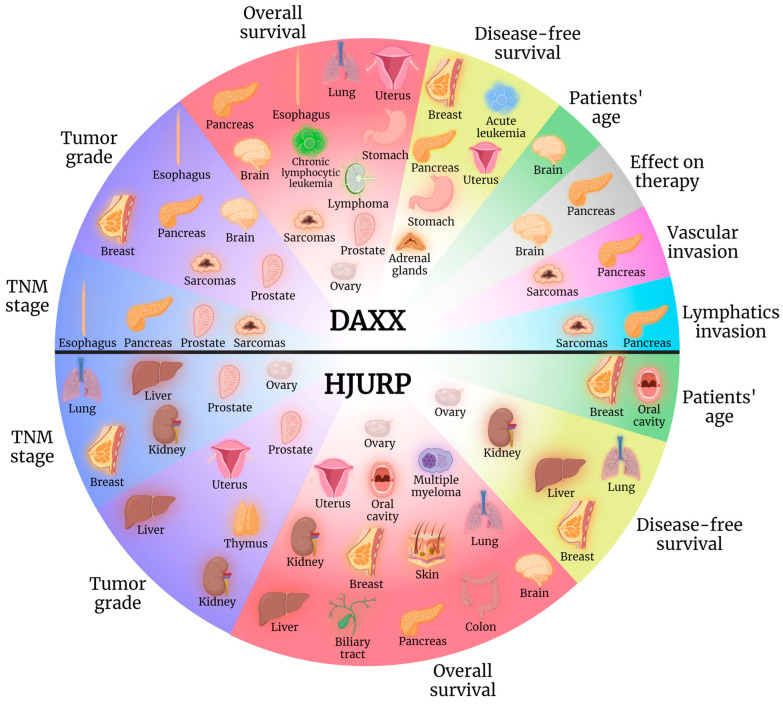
Depending on the organ of origin, the expression of DAXX and HJURP has been shown to exert an impact on a number of clinicopathological parameters, including disease TNM stage (blue field), tumor grade (purple field), and patients’ OS (red field) and DFS (yellow field). It has also been linked to patients’ age (green field) and presence of lymphovascular invasion (light blue and pink field) as well as demonstrated to affect therapeutic outcome (grey field). Created with BioRender.com.

**Table 1 cancers-15-05165-t001:** DAXX and HJURP studied in solid tumors of head and neck as well as thymus and skin and correlations with clinicopathological parameters.

Malignant Tissues	Benign Control Tissues	Methods	Results	Refs
BRAIN				
DAXX				
55 intracranial meningioma samples	-	TMA IHC	Significantly higher DAXX expression (nuclear and cytoplasmic) in grade 2/grade 3 compared to grade 1 meningiomas	[10]
74 glioblastoma multiforme (GBM) samples	-	NGS	Alterations in DAXX genes associated with Better OS. Younger age at diagnosis. Frontal or temporal tumor location. Presence of giant tumor cells.	[11]
85 GBM samples	-	qRT-PCR	Loss of DAXX expression associated with short OS	[12]
121 neuroblastomas (NBLs) in total (110 NBLs with normal or shortened telomeres and 11 NBLs with elongated telomeres by alternative lengthening of telomere (ALT))	-	NGS	No DAXX or ATRX alterations were detected in the 110 tumors with normal or shortened telomeres. DAXX mutations were detected in one case in the 11 ALT cases. ALT was caused by ATRX or DAXX alterations. ATRX or DAXX altered NBLs showed poor outcomes due to chemoresistance.	[13]
106 pituitary adenomas	-	TMA IHC Telomere-specific FISH NGS	Two out of 3 ALT-positive pituitary adenomas had alterations in either ATRX or DAXX. No significant differences in the clinicopathological characteristics of the group with abnormally shortened telomeres compared to the group with normal telomeres.	[14]
32 recurrent pituitary adenomas from 22 patients			36% of patients (*n* = 8) were ALT-positive. ALT-positive pituitary adenomas are enriched in recurrent disease.	
HJURP				
267 malignant glioma specimens	-	qRT-PCR data collection from the GEO high-grade gliomas data sets	HJURP overexpression associated with decreased OS	[15]
96 high-grade glioma samples (64 glioblastomas and 32 grade 3 gliomas)	6 adult and 4 fetal normal brain samples	IHC	Positive HJURP staining was significantly higher in glioblastomas than in grade 3 gliomas. HJURP high immunoexpression score associated with lower OS.	[16]
40 astrocytoma samples (5 diffuse astrocytomas, 5 anaplastic astrocytomas and 30 GBMs)	7 non-neoplastic white matter samples from patients undergoing temporal lobectomy for epilepsy treatment	qRT-PCR Western Blot	HJURP is Highly overexpressed in malignant gliomas. Higher HJURP expression levels associated with poor OS.	[17]
ORAL CANCER				
DAXX				
25 oral squamous cell carcinoma (OSCC) samples	Matched normal tissues from the same patients	qRT-PCR Western Blot IHC	DAXX mRNA and protein expression elevated in OSCC samples in comparison to normal tissues	[18]
HJURP				
152 oral cancer (OC) tissue specimens	Adjacent normal oral tissues	qPCR Western Blot TMA IHC	HJURP expression observed in 103 (67.8%) of 152 OC tissue specimens, but expression in normal oral tissues was limited. Positive HJURP expression was significantly correlated with ∘ Higher age. ∘ Shorter OS.	[19]
THYMUS				
HJURP				
95 thymic epithelial tumor (TET) samples	-	TMA IHC	Higher cytoplasmic HJURP expression associated with advanced Masaoka–Koga stage. HJURP cytoplasmic expression positivity associated with lymphocyte-poor TETs.	[20]
SKIN				
HJURP				
458 melanoma patients	Adjacent normal tissues	Data from GEO database retrieval	HJURP expression in melanoma was Significantly more upregulated than that observed in normal tissues. Associated with decreased patients’ OS.	[21]

ATRX, alpha-thalassemia/mental retardation, X-linked; FISH, fluorescence in situ hybridization; GEO, Gene Expression Omnibus; ICH, immunohistochemistry; NGS, next-generation sequencing; OS, overall survival; qRT-PCR, quantitative reverse transcription polymerase chain reaction; TMA, tissue microarrays.

**Table 2 cancers-15-05165-t002:** DAXX and HJURP studied in solid tumors of lung and breast as well as sarcomas and correlations with clinicopathological parameters.

Malignant Tissues	Benign Control Tissues	Methods	Results	Refs
LUNG				
DAXX				
345 lung cancer samples in total (194 samples of squamous cell lung carcinoma (SQCLC), 111 samples of pulmonary adenocarcinoma (AC), 40 samples of small cell lung cancer (SCLC))	-	IHC	High DAXX expression in: ∘ 54.9% of AC. ∘ 76.2% of SQCLC. ∘ 82.5% of SCLC. High DAXX expression associated with lower OS in AC patients, although not significantly (*p* = 0.216). High concomitant DAXX and ATRX expression associated with better OS in AC patients.	[22]
HJURP				
1090 lung cancer specimens in total (551 lung AC specimens 539 SQCLC)	-	Data collection from TCGA database	High HJURP expression is significantly associated with poor OS of lung AC patients	[23]
154 non-SCLC (NSCLC) tissue samples in total	92 paired nontumor samples in total	Data collection from the GEO database	HJURP upregulation associated with worse OS in NSCLC patients	[24]
74 NSCLC samples	adjacent normal tissue specimens	RT-PCR Western blot	HJURP expression significantly upregulated in NSCLC tissues compared to normal tissues. Increased HJURP expression was associated with ∘ Advanced TNM stage. ∘ Presence of distant metastasis. ∘ Poor OS. ∘ Lower DFS.	[25]
519 lung AC specimens	54 normal lung tissue specimens	Data collection from TCGA and the GEO databases	High HJURP expression was significantly associated with Stage. T grade. N grade. Poor OS.	[26]
BREAST				
DAXX				
220 breast cancer (BC) specimens	-	IHC	DAXX expression was significantly associated with ∘ Tumor grade. ∘ Presence of necrosis. ∘ Positive vascular invasion. ∘ Positive hormonal receptors. Low DAXX expression had a significant negative effect on DFS but not OS. Low DAXX expression was associated with a worse DFS in ER positive BC cases.	[27]
39 solid papillary carcinoma (SPC) specimens (18 SPC in situ and 21 SPC invasive)	-	IHC	Loss of DAXX expression was significantly associated with lymphatic invasion	[28]
HJURP				
44 in situ breast carcinoma specimens	-	IHC	A greater than sevenfold risk of relapse observed in patients highly expressing HJURP in stroma	[29]
351 basal-like breast cancers (BLBCs) specimens	-	Genomic and transcriptomic profile data collection from METABRIC and ROCK data sets	High levels of HJURP expression in basal II basal-like subtype (BL2) [30]. BL1 and BL2 subtypes exhibited higher expression of cell cycle and DNA damage response genes, and representative cell lines preferentially responded to cisplatin.	[31]
130 tumor specimens	-	Affymetrix Microarray	High HJURP mRNA levels were significantly associated with ER negative. PR negative. Advanced Scarff–Bloom–Richardson grade. Young age. Ki67 proliferation indices. Shorter DFS. Shorter OS.	[32]
71 BC samples	-	qRT-PCR IHC	High HJURP mRNA levels were independently prognostic for lower DFS. In the luminal A subtype, high mRNA levels of HJURP were prognostic of lower distant metastasis-free survival, higher possibility of developing metastasis and shorter OS.	[33]
SARCOMAS				
DAXX				
106 angiosarcoma specimens	-	IHC	All specimens examined retained nuclear expression of DAXX	[34]
80 chondrosarcoma specimens	25 osteochondroma specimens	IHC	Positive DAXX expression was significantly higher in chondrosarcoma than in osteochondroma tissues. Percentages of positive DAXX, DRD3, and DISC1 expression ∘ Lower in tissues with good differentiation. ∘ AJCC stage I/ II. ∘ Enneking stage I. ∘ Non-metastasis. Positive DAXX expression associated with ∘ Shorter OS. ∘ Mortality.	[35]
32 soft tissue sarcoma specimens from 14 individuals (including leiomyosarcoma (LMS), myxofibrosarcoma (MFS), rhabdomyosarcoma (RMS) and synovial sarcoma (SS))	-	Genome-wide methylation analysis	DAXX gene methylation detected in LMS specimens	[36]
92 LMS (derived from the uterus, retroperitoneum/intra-abdomen, and various other sites)	-	IHC	None of the 92 cases lost DAXX expression	[37]
92 gastrointestinal stromal tumor (GIST) specimens	-	IHC	3 of 92 cases (3.3%) showed loss of expression of DAXX. Telomere dysregulation (TERT promoter mutations or loss of either ATRX or DAXX expression) was frequently seen in GISTs of non-gastric origin but did not statistically correlate with any clinicopathological characteristics, DFS or OS.	[38]
HJURP				
58 synovial sarcoma (SS) specimens	-	Data collection on SS gene expression profile from the GEO database	HJURP gene over-expressed in SS (along with four other genes, namely *NCAPG*, *TPX2*, *CENPA*, and *NDC80*)	[39]

ATRX, alpha-thalassemia/mental retardation, X-linked; AJCC, American Joint Committee on Cancer; CENPA, centromere protein A; DFS, disease-free survival; DISC1, disrupted in schizophrenia 1; DRD3, dopamine receptor D3; ER, estrogen receptor; GEO, Gene Expression Omnibus; IHC, immunohistochemistry; NCAPG, non-SMC condensin I complex subunit G; OS, overall survival; PR, progesterone receptor; RT-PCR, reverse transcription polymerase chain reaction; TCGA, the cancer genome atlas; TERT, telomerase reverse transcriptase; TPX2, Xenopus kinesin-like protein 2.

**Table 3 cancers-15-05165-t003:** DAXX and HJURP studied in solid tumors of the gastrointestinal tract, liver, biliary tract and pancreas, and correlations with clinicopathological parameters.

Malignant Tissues	Benign Control Tissues	Methods	Results	Refs
ESOPHAGUS				
DAXX				
60 esophageal squamous cell carcinoma (ESCC) specimens	Adjacent non-tumorous tissues	IHC	DAXX expression (higher percentage of cells) was associated with ∘ N stage. ∘ OS in stage IIB and stage IV disease. ∘ AJCC stage. ∘ Presence of distant metastasis. Non-neoplastic squamous epithelium adjacent to the cancer showed negative or weak staining.	[40]
81 ESCC and 19 paired ESCC LN metastases specimens	31 paired non-neoplastic esophageal epithelia	TMA IHC	Nuclear expression of DAXX was significantly correlated with nuclear DJ-1 expression. DJ-1 expression was significantly higher in ESCC and ESCC lymph node metastases than in nonneoplastic esophageal epithelium. ESCC specimens with high distant metastatic potential (regarded as having a high distant metastatic potential if distant metastasis could be detected within 1 year of esophagectomy; else, they were considered to have a low distant metastatic potential) also had a significantly higher level of nuclear DJ-1 expression. Nuclear DJ-1 is an independent predictor of patient OS, associated with lower patients’ OS.	[41]
STOMACH				
DAXX				
323 gastric cancer (GC) specimens	20 paired adjacent normal tissues	TMA IHC	DAXX expression positive rate was higher in tumor tissues (65%) than in adjacent normal tissues (50%). DAXX is highly expressed in GC and exerts opposite effects in different subcellular locations. Cytoplasmic DAXX expression was associated with a longer OS and DFS, while high nuclear DAXX expression suggested a poorer OS.	[42]
70 GC specimens	70 paired adjacent normal tissue specimens	IHC	DAXX Nuclear/Cytoplasmic ratio (NCR) significantly higher than that in the tissues adjacent to the carcinoma. In normal gastric mucosal cells, DAXX was clearly expressed in the cytoplasm but was absent in the nucleus. Intestinal metaplasia cells appeared to express DAXX in both the nucleus and the cytoplasm, and in tumor cells, DAXX was clearly expressed in the nucleus but was absent in the cytoplasm. High DAXX NCR ratio was associated with shorter DFS and OS.	[43]
522 GC specimens	-	TMA IHC		
COLON				
DAXX				
106 colorectal carcinoma (CRC) samples	Matched nontumor-surrounding tissues	Western blot	DAXX expression significantly lower in the patients with positive serum CEA screening results (defined as serum CEA levels of >5 ng/mL) compared to patients with negative CEA screening levels. DAXX expression was significantly correlated with CD24 expression. Reduced DAXX expression was associated with reduced CD24 expression in CRC tissues.	[44]
8 pairs of matched primary CRC tissue and liver metastatic CRC tissue specimens	-	Real-time RT-PCR IHC	DAXX expression lower in liver metastases than in primary CRC tissues	[45]
8 CRC specimens	Matched adjacent tissue specimens	Western blot	DAXX expression reduced in CRC tissues compared to normal colon tissues	[46]
28 CRC specimens	Adjacent normal tissues	Data collection from TCGA data sets	DAXX expression increased in 19 of the 28 CRC specimens, compared with the adjacent normal tissues	[47]
15 CRC specimens	Adjacent normal tissues	qRT-PCR IHC	DAXX RNA expression increased in 12 of the 15 CRC specimens, compared with the adjacent normal tissues, in quantitative RT-PCR studies. DAXX expression elevated in the nuclei of the adenocarcinoma cells compared to that of the adjacent normal intestinal epithelium in IHC studies.	
HJURP				
162 CRC specimens	-	IHC	High HJURP expression detected in 44 CRC tissues (27.2%) and low in 118 CRC tissues (82.8%). High HJURP expression was associated with higher patients’ OS.	[48]
LIVER				
HJURP				
621 hepatocellular carcinoma (HCC) specimens in total	292 non-tumor specimens	Data collection from TCGA and the GEO datasets	HJURP mRNA expression was significantly associated with HCC patients’ OS. Invasion depth of tumor. Distant metastasis.	[49]
176 HCC specimens	21 adjacent normal tissue specimens	Real-time PCR IHC Data collection from the GEO, TCGA and the ICGC databases	HJURP expression ∘ Higher in HCC tissues compared to adjacent normal tissues. ∘ Higher in tissues of patients with advanced HCCs than those of patients with early-stage disease. HJURP overexpression was associated with ∘ Poor DFS. ∘ Poor OS.	[50]
164 HCC specimens	-	IHC real-time PCR	High HJURP expression was significantly associated with Tumor size (in tumors >5 cm). Barcelona clinic liver cancer stage. Tumor number. Tumor differentiation. TNM stage. Poor OS.	[51]
BILIARY TRACT				
HJURP				
127 Cholangiocarcinoma (CCA) specimens in total, including 32 intrahepatic (iCCA), 71 perihilar (pCCA) and 24 distal (dCCA) CCA specimens	-	TMA IHC	HJURP overexpression was associated with lower OS of iCCA and pCCA patients but not in dCCA patients	[52]
-	-	Data collection from TCGA database	HJURP expression upregulated in CCAs compared with adjacent non-tumorous tissues	
PANCREAS				
DAXX				
68 sporadic pancreatic neuroendocrine tumors (PanNETs) specimens (that were not part of a familial syndrome associated with PanNETs)	-	Whole-exome sequencing	Mutations of DAXX genes were detected in 12 out of 68 cases and associated with better OS	[53]
1322 NETs in total 561 primary nonfunctional pancreatic neuroendocrine tumours (NF-PanNETs), 107 NF-PanNET metastases and 654 primary, non-pancreatic non-functional NETs and NET metastases	-	IHC	The loss of DAXX was correlated with distant metastases/recurrence. DFS rates were shorter in patients with small NF-PanNETs (≤2.0 cm) with DAXX loss. Only a small subset of pulmonary carcinoids exhibited DAXX loss. Therefore, the (loss of) DAXX is a highly specific diagnostic biomarker to indicate the possibility of a pancreatic primary.	[54]
44 PanNET specimens	-	IHC	Low DAXX expression was correlated with Nonfunctional tumors. Higher Ki-67 index. WHO grade G2. Higher recurrence rate. Worse DFS. Presence of synchronous liver metastasis, although insignificantly (*p* = 0.081).	[55]
13 pancreatic neuroendocrine neoplasm (NEN) specimens (4 G1, 5 G2 and 4 G3 tumor samples)	Matched non-tumor tissues	qRT-PCR IHC	DAXX had lower mRNA expression in the tumor relative to that in the non-tumor tissue in G1 and G2 specimens. In contrast, G3 samples exhibited higher DAXX mRNA expression in tumor tissues relative to non-tumorous ones. DAXX IHC staining was absent in non-tumor and G1 and G2 tumor tissues, while strong nuclear staining was observed in less than 25% of cells in G3 tumor tissues.	[56]
51 neuroendocrine liver metastases specimens	-	NGS	DAXX mutation identified in 13 out of 51 cases (25%). DAXX mutation was associated with shorter hepatic progression-free survival. DAXX mutation status was concordant between primary and metastatic sites.	[57]
76 PanNET specimens	-	TMA IHC	Loss of DAXX protein expression was associated with Poor OS. Longer OS in the subgroup of metastatic PanNETs.	[58]
243 well-differentiated (G1 and G2) primary PanNET specimens	-	TMA IHC	The loss of DAXX was associated with Tumor stage. Presence of metastasis. Reduced DFS. Decreased OS.	[59]
53 PanNET specimens (46 primaries and 7 liver metastases from 39 cases)	-	IHC	The loss of DAXX expression was associated with poorer 5-year DFS	[60]
37 PanNET specimens	-	Sanger sequencing	DAXX/ATRX mutations were Identified in 54.05% of samples. Associated with shorter OS.	[61]
PanNET specimens from patients with Von Hippel-Lindau (VHL) syndrome	-	IHC	No statistically significant difference in tumor DAXX protein expression via pathologic diagnosis, clinical data, VHL exon mutation, and functional imaging results was identified.	[62]
16 PanNET specimens	-	IHC	DAXX loss was associated with Postoperative hepatic recurrence. Lower DFS.	[63]
87 small primary PanNET (<3 cm) specimens	-	IHC NGS	The presence of DAXX mutation was associated with an increased risk of metastasis.	[64]
269 primary PanNET specimens	19 sporadic microadenomas	TMA IHC	The loss of ATRX/DAXX nuclear expression was observed in 19.3% of tumors but not in microadenomas. The loss of ATRX/DAXX expression was associated with ∘ ALT. ∘ Reduced DFS. However, in patients with distant metastases, the loss of ATRX/DAXX expression was associated with better OS.	[65]
100 Grade 1 and 2 primary PanNET specimens	-	IHC	The loss of DAXX was detected in 64 out of 100 cases.	[66]
105 PanNET specimens	-	IHC	The loss of DAXX expression was detected in 16/105 samples (15.2%). The loss of DAXX expression alone did not significantly correlate with poor OS. The loss of either DAXX or ATRX was associated with ∘ Higher Ki67 proliferative index. ∘ High grade. ∘ Nodal involvement. ∘ Higher stage. ∘ Infiltrative borders. ∘ Lymphovascular invasion.	[67]
46 PanNET specimens	-	IHC	The loss of ATRX/DAXX expression was Detected in 18/46 cases (39.1%). Associated with tumor size larger than 5 cm.	[68]
164 tumor specimens in total (10 gastric, 15 duodenal, 20 rectal, 70 pancreatic, and 22 pulmonary NETs and 27 pancreatic adenocarcinomas (PDACs))	15 nonneoplastic pancreas specimens	IHC	The loss of DAXX or ATRX expression was detected in at least 1 case of NETs from all organs examined but not in any of nonneoplastic pancreatic tissues or PDACs. The loss of DAXX expression detected in 29.9% of all NET cases. The loss of DAXX protein expression was associated with ∘ Ki-67 index. ∘ Higher histologic grades. Statistical analysis was performed separately in different primary organs did not result in significant associations between DAXX expression and Ki-67 index or tumor grade.	[69]
HJURP				
177 PDAC specimens	Adjacent normal tissues	Data collection from the GEO dataset	HJURP expression was Significantly higher in PDAC tissues than in adjacent normal tissues. Associated with poor OS.	[70]
219 PDAC specimens	Adjacent normal tissues	IHC qRT-PCR Western blot		

AJCC, American Joint Committee on Cancer; ALT, alternative lengthening of telomere; ATRX, alpha-thalassemia/mental retardation, X-linked; CEA, carcinoembryonic antigen; DFS, disease-free survival; GEO, Gene Expression Omnibus; ICGC, International Cancer Genome Consortium; IHC, immunohistochemistry; LN, lymph node; OS, overall; qRT-PCR, quantitative reverse transcription polymerase chain reaction; TCGA, the cancer genome atlas; TMA, tissue microarrays; WHO, World Health Organization.

**Table 4 cancers-15-05165-t004:** DAXX and HJURP studied in solid tumors of the kidney, bladder, prostate, and testis and correlations with clinicopathological parameters.

Malignant Tissues	Benign Control Tissues	Methods	Results	Refs
KIDNEY				
DAXX				
11 primary renal well-differentiated neuroendocrine tumors (NET) specimens	-	NGS	No DAXX mutations were detected in any of the cases	[71]
HJURP				
416 clear cell renal cell carcinoma (ccRCC) specimens	-	Data collection from TCGA dataset	HJURP overexpression was associated with Reduced DFS. Poorer OS.	[72]
52 ccRCC specimens	Adjacent normal tissues	Data collection from TCGA dataset	No association between HJURP expression and clinicopathological parameters observed	[73]
15 ccRCC specimens	Adjacent paracancerous renal tissue samples	qRT-PCR Western blot	HJURP expression lower in ccRCC tissues compared with that in the adjacent paracancerous ones	[74]
539 ccRCC specimens	72 paracancerous normal tissue samples	Data collection from TCGA dataset	HJURP expression upregulated in ccRCC tissues compared to normal ones HJURP expression was associated with ∘ T stage. ∘ N stage. ∘ M stage. ∘ Clinical stage. ∘ Histopathological grade. ∘ Poor OS. HJURP levels of expression correlated with infiltration of various immune cells and expression of a wide range of immunocyte gene markers	[75]
BLADDER				
DAXX				
5 pT1 bladder urothelial carcinoma (UC) specimens	5 normal bladder urothelium samples	IHC	DAXX expression observed in 90% of UC nuclei compared to 70% of normal urothelium nuclei. In UC tissues, DAXX expression in nuclei adjacent to the stroma is slightly lower than in the intermediate cell layers, whereas in normal urothelium tissues it is slightly greater compared to the intermediate and superficial cell layers	[76]
31 small cell bladder carcinoma specimens	-	NGS	loss of DAXX expression observed in 5 (16.1%) cases, including 4 patients who presented with limited stage disease. All 4 patients who presented with limited-stage disease underwent chemotherapy and remained alive during follow-up, implying that loss of DAXX may sensitize tumor cells to chemotherapy.	[77]
PROSTATE				
DAXX				
115 prostate cancer specimens, 8 prostatic intraepithelial neoplasia specimens	37 benign prostatic hyperplasia samples	TMA IHC	Higher DAXX expression observed in prostate cancer tissues. DAXX overexpression was associated with ∘ Gleason score. ∘ Presence of metastasis.	[78]
50 prostate cancer tissue specimens	50 normal adjacent tissue samples from the same cases with prostate cancer 50 benign prostatic hyperplasia (BPH) tissue samples	qRT-PCR	DAXX expression significantly increased compared to both control groups (normal adjacent and BPH tissues)	[79]
5718 prostate cancer specimens	Normal prostate tissues	TMA IHC	DAXX expression detected in 4609 (80.6%) of 5718 cases High DAXX expression was associated with ∘ TMPRSS2/ERG rearrangement. ∘ ERG expression. ∘ High Gleason grade. ∘ Advanced pT stage. ∘ High Ki-67 labeling index. ∘ Early prostate-specific antigen recurrence.	[80]
Prostate cancer tissues	Normal prostate tissues	Data collection on DAXX mRNA expression levels from the Oncomine database	DAXX mRNA levels increased in metastatic prostate cancer tissues but low in normal prostate glands. Increased DAXX expression was associated with ∘ Lower OS. ∘ Higher Gleason scores.	[81]
HJURP				
99 prostate cancer specimens	81 benign prostate tissue samples	TMA IHC	HJURP overexpressed prostate cancer tissues compared with benign prostate tissues. High HJURP expression was associated with ∘ Positive PSA levels. ∘ High Gleason score. ∘ Advanced pathological stage. ∘ Metastasis. ∘ PSA failure. ∘ Shorter biochemical recurrence-free survival.	[82]
22 prostate cancer specimens	10 benign prostate tissue samples	qRT-PCR		
484 prostate cancer specimens	-	Data collection from TCGA dataset	High HJURP expression (along with high expression of UBE2C, PLK1, CDC20, BUB1 and CDK1 genes) was associated with worse biochemical recurrence-free survival	[83]
ADRENAL GLANDS				
DAXX				
43 adrenal cortical carcinoma specimens	50 cortical adenoma samples	TMA IHC	Global loss of DAXX expression was associated with longer DFS	[84]

BUB1, budding uninhibited by benzimidazoles 1; CDC20, cell division cycle protein 20 homolog; CDK1, cyclin-dependent kinase 1; DFS, disease-free survival; IHC, immunohistochemistry; NGS, next-generation sequencing; OS, overall survival; PLK1, polo-like kinase 1; PSA, prostate-specific antigen; qRT-PCR, quantitative reverse transcription polymerase chain reaction; TCGA, the cancer genome atlas; TMA, tissue microarrays; TMPRSS2, transmembrane protease, serine 2; UBE2C; ubiquitin conjugating enzyme E2 C.

**Table 5 cancers-15-05165-t005:** DAXX and HJURP studied in solid tumors of the ovaries, endometrium, and cervix and correlations with clinicopathological parameters.

Malignant Tissues	Benign Control Tissues	Methods	Results	Refs
OVARY				
DAXX				
59 tumor specimens (36 serous cystadenoma, 13 endometrioid carcinoma, 3 clear cell carcinoma, 4 mucinous cystadenoma, and 3 granular cell tumor samples)	3 normal ovary tissue samples	IHC	High DAXX staining intensity was identified in ∘ Endometrioid carcinoma (92.3%). ∘ Serous cystadenoma (91.7%). ∘ Clear cell carcinoma (100%). ∘ Mucinous cystadenoma (100%). ∘ Granular cell tumor tissues (0%). Therefore, DAXX was highly expressed in human ovarian surface epithelial tumors but not in granulosa cell tumors and weakly expressed in normal ovarian tissues.	[8]
400 high-grade serous carcinoma serous effusion specimens	-	IHC	DAXX expression was detected in 348/400 (87%) of cases with IHC. DAXX expression via IHC was observed to be higher ∘ In pleural compared to peritoneal effusions. ∘ In post-chemotherapy compared to pre-chemotherapy effusions. DAXX expression was detected in 70/81 (86%) of cases with Western blotting. DAXX expression was associated with poor OS in the entire cohort.	[85]
81 of the 400 high-grade serous carcinoma effusion specimens	-	Western blot		
187 epithelial ovarian cancer specimens	-	qRT-PCR	No statistically significant association between DAXX mRNA expression and clinicopathological parameters was observed	[86]
HJURP				
156 ovarian cancer specimens	Fallopian tube tissue samples from patients undergoing salpingectomy for benign diseases	TMA IHC	High HJURP expression was observed in 86/156 (55.13%) tumor specimens, as opposed to 22/74 (29.73%) fallopian tube samples. High HJURP expression was associated with ∘ High (≥200 U/mL) serum carbohydrate antigen 125 (CA125) level. ∘ Large (≥3000 mL) ascites volume. ∘ Omentum/peritoneum metastasis. ∘ Lower DFS. ∘ Lower OS.	[87]
98 advanced-stage (stage IIIB–IV) serous ovarian carcinoma specimens	-	IHC	Low HJURP expression levels were detected in 66.33% (65/98) and high levels in 33.67% (33/98) of cases. High HJURP expression levels were significantly associated with ∘ Lymph node metastases. ∘ Lower OS.	[88]
UTERUS				
DAXX				
Endometrial clear cell carcinomas specimens	-	NGS	DAXX homozygous deletions were detected on 11% of cases	[89]
18 Uterine smooth muscle tumours of uncertain malignant potential (STUMP) and 43 leiomyosarcoma specimens	-	IHC	The loss of ATRX or DAXX expression was observed in all uterine STUMP (4/4) and vaginal STUMP (2/2) patients and almost all leiomyosarcoma patients (88.4%, 23/26, including 90% (9/10) of stage 1 leiomyosarcoma cases) who had died of disease or who had recurrent disease. The loss of ATRX or DAXX expression in leiomyosarcoma patients was associated with ∘ Lower DFS. ∘ Lower OS.	[90]
142 various uterine leiomyoma (UL) subtype specimens (consisting of 35 cellular ULs, 45 highly cellular ULs, 31 mitotically active ULs (including 15 tumors also displaying cellularity (*n* = 7) or high cellularity (*n* = 8) and 31 ULs with bizarre nuclei) and 64 conventional UL specimens	-	IHC	The loss of DAXX expression was observed only in 1 case of cellular UL	[91]
HJURP				
552 endometrial carcinoma specimens	23 normal endometrium samples	Data collection from the GEO dataset	High HJURP mRNA expression was associated with High tumor grade. Recurrence. Worse OS.	[92]
CERVIX				
DAXX				
Cervical intraepithelial neoplasia grade 1 (CIN1), CIN2, CIN3, and cervical cancer specimens	Normal cervical epithelial tissue samples	IHC	DAXX expression was located In the nucleus of normal cervical epithelial cells. Nuclear membrane and in a lesser extent the nucleus of CIN1 cells. In the cytoplasm and cell membrane in CIN2, CIN3, and cervical cancer cells.	[93]

ATRX, alpha-thalassemia/mental retardation, X-linked; DFS, disease-free survival; GEO, Gene Expression Omnibus; IHC, immunohistochemistry; OS, overall survival; qRT-PCR, quantitative reverse transcription polymerase chain reaction; TMA, tumor microarrays.

**Table 6 cancers-15-05165-t006:** DAXX and HJURP studied in hematological malignancies and correlations with clinicopathological parameters.

Malignant Tissues	Control Tissues	Methods	Results	Refs
LEUKEMIA				
DAXX				
100 BM specimens of pediatric acute leukemia (AL) (49 acute lymphoblastic leukemia (ALL), 21 acute non-lymhocytic leukemia (ANLL), 30 acute myeloid leukemia (AML))	30 BM specimens of healthy children	IHC	Increased rate of DAXX protein expression compared to controls ALL 30.38%. ANLL 66.67%. AML 46.67%. Healthy controls 6.67%.	[94]
50 BM specimens of pediatric AL (34 ALL, 16 ANLL)	20 BM specimens of healthy children	IHC	Increased rate of DAXX protein expression ANLL 62.5%. ALL 26.5%. HR ALL 55.6.	[95]
50 BM specimens of pediatric AL	20 BM specimens of healthy children	IHC	Increased rate of DAXX protein expression All AL 38%. HR ALL 55.6%.	[96]
BM specimens of 8 AML cases at diagnosis and relapse	-	WGS	Recurrent somatic DAXX mutations in relapsed AML cases	[97]
MYELODYSPLASTIC NEOPLASMS				
DAXX				
77 BM aspiration specimens	Control CD34+ BM cells	Real-time PCR	Increased DAXX mRNA expression in myelodysplastic neoplasms (MDS) CD34+ cells	[98]
CHRONIC LYMPHOCYTIC LEUKEMIA				
DAXX				
PB and BM aspiration specimens from 62 untreated patients	-	Flow cytometry	Positive correlation between Par-4 and DAXX protein expression	[99]
LYMPHOMA				
DAXX				
Purified B cells derived from cell suspensions of lymph nodes from 5 diffuse large B cell lymphoma (DLBCL) patients	Purified B cells derived from cell suspensions of tonsils from 2 controls	Western blot	Ιncreased DAXX protein expression in DLBCL B cells compared to tonsil B cells	[100]
196 DLBCL tissues	76 follicular lymphoma (FL) tissues, 52 chronic lymphocytic leukemia (CLL) tissues	Tissue microarray analysis	Increased DAXX protein expression in DLBCL compared to FL and CLL. DAXX expression was correlated with poor OS for all DLBCL subtypes.	
93 biopsy samples of newly diagnosed natural killer T cell lymphoma (NKTCL)	-	IHC	DAXX protein overexpression in HEA subtype. DAXX expression was correlated with NF-κB and TCR pathways.	[101]
MULTIPLE MYELOMA				
HJURP				
GSE2113: purified plasma cells from 7 nonoclonal gammopathy of undetermined significance (MGUS), 39 newly diagnosed multiple myeloma (MM) and 6 plasma cell leukemia (PCL) patients. GSE6477: 162 samples from MGUS, smoledringMM, newly diagnosed MM, refractory MM.	-	Gene Expression Omnibus datasets: Expression profiling by array	Gradual increase in *HJURP* expression along with disease progression from MGUS to PCL. Higher *HJURP* expression in rMM compared to non-rMM.	[102]
MM patient datasets (CoMMpass, HOVON, and UAMS)	-	WGS	High *HJURP* value was associated with worse OS and PFS	

BM, bone marrow; HEA, HDAC9, EP300, and ARID1A mutations; HR, high risk; IHC, immunohistochemistry; NF-Κb, nuclear factor kappa B; OS, overall survival; Par-4, prostate apoptosis response-4 protein; PB, peripheral blood; PFS, progression-free survival; TCR, T cell receptor; WGS, whole-genome sequencing.

## Data Availability

Not applicable.

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
