# Peer review of "The Clinical Impact of Death Domain-Associated Protein and Holliday Junction Recognition Protein Expression in Cancer: Unmasking the Driving Forces of Neoplasia"

_cancers, 2023, doi:10.3390/cancers15215165_

Round 1

Reviewer 1 Report (Previous Reviewer 1)

The Author have addressed all the comments properly, the article should be accepted

Author Response

We would like to sincerely thank the reviewer for his kind comments as well as his previous suggestions that enabled us to improve our manuscript.

Reviewer 2 Report (Previous Reviewer 2)

The manuscript did an extensive literature review and summarized results from studies investigating the association between DAXX and HJURP expression in human 74 tumor specimens and clinicopathological parameters.  The author has addressed some of my previous comments by adding study selection criteria in section 1. However, this manuscript is still limited to summarizing the evidence from previous studies. Interpretation of the results and adding their perspectives/thoughts are lacking. For example, it was mentioned in Section 8 line 1083 that “contradictory data have been reported regarding the 1083 properties of DAXX and HJURP in neoplasia“. The authors didn’t present perspectives or critical analysis on why contradictory data have been reported. 

Author Response

We would like to thank the reviewer for his helpful feedback and suggestions.

The following text has been added to the section “Conclusion” discussing our thoughts on the fact that some studies reported contradictory data regarding the tumor-promoting or tumor-suppressing role of DAXX and HJURP in the same tumor types.

“This can be attributed to the aberrant amplification of different molecular pathways leading to carcinogenesis. Each tumor carries a unique molecular signature and deeper understanding of it comprises a key step for the development of personalized therapeutic interventions. Nevertheless, such discrepancies render evident the need for further research to elucidate the complex role of DAXX and HJURP in carcinogenesis. Especially for tumors of the same histological type, where some research teams detected tumor-promoting and others tumor-suppressing properties of the aforementioned molecules, investigation of their expression and its impact in large series of tumor samples is called for.”

This manuscript is a resubmission of an earlier submission. The following is a list of the peer review reports and author responses from that submission.

Round 1

Reviewer 1 Report

The articleThe clinical impact of DAXX and HJURP expression in cancer: 2 unmasking the driving forces of neoplasia

In the present study, presented the data reported from studies investigating 18 the clinical impact of DAXX and HJURP expression in tumors of various origins and have 19 associated the expression of DAXX and HJURP with a multitude of clinicopathological parameters, 20 including disease stage, tumor grade, patients’ overall and disease-free survival as well as lympho- 21 vascular invasion, rendering them as possible targets of future, personalized and precise therapeutic 22 interventions.

The data is nicely presented. Authors need to address following comments

Minor comments:

  1. Abstract must be modified and written in detail
  2. English language needs improvement.
  3. Kindly cite recent literature.
  4. Was consent form obtained?
  5. Figure legends should be explained in detail.

Minor modifications 

Reviewer 2 Report

The manuscript did an extensive literature review and summarized results reported from studies investigating the association between DAXX and HJURP expression in human tumor specimens and clinicopathological parameters.  

1.     It is unclear how and how many of those studies were selected and why they are representative to be included in your analysis.

2.     It was mentioned in Section 6 line 303 that “Researchers that conducted studies on HJURP expression in clear cell renal cell carcinoma (ccRCC) specimens reported contradictory results regarding its role in tumorigenesis“. The author didn’t add thoughts on why contradictory data have been reported.

Reviewer 3 Report

The authors have presented a comprehensive review of the clinical studies which have identified DAXX or HJURP in clinical samples. However, it is unclear whether this manuscript is submitted as a narrative review or a systematic review. Unfortunately, based on the guidelines outlined in the instructions to authors on the Cancers Website, I do not feel it has reaches the required standard for either review type. For the most part the manuscript is limited to presenting summaries that merely restate information from published reports. The critical component of narrative reviews is a comprehensive narrative synthesis of evidence extracted from the existing literature, which is currently missing from this manuscript. Equally, if this is a systematic review it does not follow the PRISMA guidelines and doesn’t critical analyse or drawn new conclusions from the data presented.

The authors have clearly done lot of work gathering all the studies presented in the manuscript, however they need to go one step further in the analysis and critical evaluation to draw their conclusions or present a novel perspective that is going to aid progress in their research field.

Additionally, in figures 2 & 3, the take home message the authors want the reader to take away from these images is not clear, with modification these could become more impactful and strengthen the article.